# Analysis Methodology for Evaluation of Time-Delay Impact on Network-Based System for Droop-Controlled AC Microgrid

**Yao Zhang**

School of Automation, Hangzhou Dianzi University, Hangzhou 310018, China; yaozhang@hdu.edu.cn;
Tel.: +86-571-8771-3513

**Abstract:** Potential performance improvements can be obtained by introducing communication techniques in the power electronic systems. However, network-induced time-delay could bring negative consequences of degrading performance or destabilizing the system in many cases. To investigate and handle the impacts of time-delay, a suit of analytical methodology is proposed, where both delay-insensitive and delay-sensitive control strategies of network-based system have their theoretical methods and different problem-solving paths. The former is to predict the maximum allowable boundaries of time-delay for releasing more network resources, and the latter is to use the controller-altering method for changing its original instability, respectively. The proposed methodology is concretely applied in network-based system of droop-controlled AC (Alternating Current) microgrid in islanded mode. Two different current-sharing strategies are mathematically analyzed and given to verify their validity. Experimental results also show the effectiveness of the proposed methodology in droop-controlled AC microgrid system, which provides theoretical guidance on how to use network-based control for the other power electronic systems.

**Keywords:** Network-based control; feedback gain; time-varying delays; droop-controlled AC microgrid

---

## 1. Introduction

The communication technique has become increasingly important in power electronics area with its developing tendency of standardization and integration. Accordingly, the network-based approaches, which could be implemented for the function extension and performance optimization, are adopted in power electronic systems.

In general, two approaches can increase the capability of network-based control system effectively in power electronic systems. One is the proper design of the suitable network management and network interface, and the other is the rational implementation of control strategy and transmission allocation of signals. For example, in order to realize the synchronization function among multi-inverters, PWM (Pulse Width Modulation) period counter value or synchronized pulse [1] were allocated as transmission signals. Occasionally, not only single data, but also multi-variable may be sent simultaneously via multicast transmission. Paper [2] introduced a control method in microgrid, where grid current $i_{grid}$, grid voltage amplitude $v_{grid\_pk}$ and phase $v_{grid\_\theta}$ are packed into CAN (Controller Area Network) bus to facilitate the output quality and smooth transition from grid-connected to islanded mode. In this sense, in order to expand communication capacity to adapt the increasingly transmission speed and various signal type, some significant efforts had been done. Double-interface with USB and CAN was developed in wind-turbo power electronic system [3], or the wireless networks with radio frequency were tried in speed control of induction motor and spatial

distribution DC-DC parallel systems [4,5]. The possible communication standard and interface for the future microgrid were outlined in previous work [6]. Although they could bring some advantages like high transmission speed, other issues like high hardware cost and severe electromagnetic interference still hinder its wide utilization. It is more realistic to take full advantage of currently available communication facilities.

Besides the extensive use of network, network-based control can also be utilized to further improve the performance of power electronic systems. However, the use of communication networks usually introduces some challenging issues. For example, the limited bit rate of communication networks would lead to network-induced delays or packet dropouts. These issues are generally regarded as sources degrading system performance or destabilizing the system, even though some positive effects of network-induced delays exist in network-based system [7]. Since it might be complicated to update the advanced communications for decreasing the delays, we may embark on doing some realistic work to adjust changing network conditions, such as altering transmission and data dispatching modes, or modifying the control strategies of power electronic converters. Paper [8] proposed two algorithms to decrease the impact of transmission time-delay on sensor signals. Time-delay is regarded as constant by setting synchronization trigger signal and time-table for receiving plant. Another method is to estimate sensor data and next sampling time. In order to compensate the effect of time-delay, some strategies are frequently employed, for instant changing PWM carrier-wave or time offset, as reported by Reference [9]. In this case, it is inevitable to change the hardware as timer clock or communication interface with the aid of software, as introduced in a three-phase Boost converter system using PEBB (Power Electronics Building Block) communication standard [10]. In those techniques, the saving time-delay may not be worth the effort or cost having been put in the hardware and software.

A traditional step is to predict the effect and build the mathematical model with the time-delay [11,12]. For power electronic converters, the stability analysis with communication time-delay tends to be investigated based on switching period, the solution of stability function is obtained from different switch states under time-delays in the same time axis [9]. The research was extended to a DC/DC parallel operation with wireless communication. Through using the time-delay stationary model and its related restriction conditions, the stability can be determined through state average model of closed-loop [5]. Other work focused on the calculation of stability boundary, piecewise linear model with time-delay is one of the common methods [4]. The direct way is to achieve from experiments and simulations, observe the output results under different time-delay [13,14]. However, the application of these stability analysis to power electronic systems is always based on relatively constant time-delay, time-varying delay is rarely considered. Although time-delay sometimes is detrimental to the stability and performance of the control system, allowing higher allowable time delay leads to efficient utilization of communication network's resources [4]. How to achieve optimal compromise between performance improvement and allowable communication speed becomes challenging in network-based power electronic systems. In addition, there are few reports about systematic analysis of time-delay impact on power electronic systems in the previous work.

With the increased concerns on environment, more and more renewable energy sources, such as wind turbines and photovoltaic cells, are integrated into the grid and formed distributed generation (DG) units. These DGs are normally interfaced to the grid with power electronic converters such as DC-AC inverters. To overcome some problems, such as system resonance and protection interfaces, the concept of AC microgrid is presented to realize the flexible coordinated control among DG units. The microgrid can operate in either grid-connected mode or islanded mode. In islanded AC microgrid, load demand must be properly shared among DG units. On some occasions, this topic is probably turned into inverter parallel operation. Conventionally, the frequency and voltage magnitude droop control method are employed, which can obtain AC microgrid power sharing in a decentralized manner [15,16]. Many advanced methods based on droop control had been reported for inverter parallel operation [17–19]. For example, VPD/FQB droop control is proposed in to offer an improved performance for controlling low-voltage AC microgrid with highly resistive transmission

lines [20]. However conventional droop-based control method has the problems of slow transient response, line impedance dependency, and inaccurate power-sharing. Some strategies were accordingly presented as Complex Line Impedance-Based Droop Method [21], Angle Droop Control [22], and Voltage-Based Droop Control [23]. However, because not all issues can be settled by only one solution, the virtual impedance methods [24] were proposed to further enhance current-sharing performance. Those methods for the parallel connected inverters were trying to avoid the dependency on communications. However, the load sharing of droop method is not good as that of methods with interconnections, and the simplicity inhered in the conventional droop method is invalid since many complex algorithms of droop method were presented. Network-based control schemes were proved to be effective for optimizing the performance of microgrids under high-penetration level of DG resources [13,25] are being used in inverter parallel operation as well. More accurate power sharing, higher reliability, and robustness can thereby be achieved [14]. To obtain superior current-sharing performance both in steady and dynamic state, network-based control on droop-based method would be a good alternative. However, the possibly negative effect of time-delay generated by network transmission cannot be neglected, therefore it goes back to the original issue, i.e., how to estimate the quantity of time-delay and estimate its influence on system stability or performance, which is seldom considered in primary droop-based control in hierarchical AC microgrid. Recently, there are reports about how to use distributed control in microgrids [26–28], which were focused more on secondary control, trying to decrease the communication links to achieve full-distributed manner. There is still a lot of research space we are required to fill using network-based strategies in primary control to better improve performances in microgrids, where communication techniques could be integrated into secondary control.

Inspired by the idea of weighting method and comprehensive evaluation methodology as mentioned in Reference [29], a suit of analytical methodology for investigating the impact of network-induced time-delay on network-based power electronic system is proposed in this paper. Inverter-parallel operation in AC microgrid working with islanded is specified as research application, and two network-based control strategies are introduced for clarifying the relationship between current-sharing performance and network-induced time-delay. The analytical methodology toward the impact of time-varying delay is described as two scenarios, which is releasing more allowable time-delay as possible while keeping system satisfactorily stable and well-functioning and altering the feedback controller based on communication status to reverse its original instability.

This paper is organized as follows. Section 2 provides a description of analytical methodology for evaluating impact of time-delay on system. Section 3 proposes two network-based control methods for the multi-inverter parallel operation in islanded mode of AC microgrids and analyzes their sensitivity toward time-delay in regard to system stability. Section 4 addresses the analytic approaches towards how to handle the time-delay, and the two current-sharing methods are contrasted to specify the methodology from the mathematical angle. Experimental results of the proposed analysis are given in Section 5. Section 6 concludes the main contribution of this paper.

## 2. Analysis Methodology for Impact of Time-Delay on Performance in Power Electronic System

We may have problems or challengers when we take advantage of the convenience of the communications in power electronic field. For example, the potentially negative aspects, such as time-delay and data dropouts, would somehow hazard the network-based control system or degrade the performance. The flow diagram shown in Figure 1 includes an analysis methodology that explains how to deal with the negative effect of time-delay.

First, the "Preliminary Analysis" as indicated in Figure 1 needs to be investigated. In a theoretical way, the mathematical model of power electronics system without considering network could be built in the form of discrete or continuous functions, by means of state-space averaging method, or circuit averaging method, etc. To achieve the desired accuracy, model may be necessary built with sampling period being several times switching cycles, which inherits the dynamically rapidity

feature of power electronic systems. Then, the time-delay is introduced in models and how the model becomes if time-delay equal to or less than one sampling period can be observed theoretically. Since the model is transformed into discrete-time one with constant time-delay, the upper limit value of time-delay whilst guaranteeing system stable can be calculated by means of stability analysis criterion on time-invariant system.

Subsequently, a judgement based on the upper limit value of time-delay is loaded. If the upper limit value is less than one sampling period (marked as "Y"), that means the network-based control system is highly sensitive to the time-delay. As a result, we may boost the transmission rate of network. whereas additional network load is dramatically increased. In this case, we may need to study the alternative approaches to alter stability condition with given time-delay. The work could be started on the revision of data allocation method, network management, and so on. A potential solution to reverse its instability is to redesign the controller based on communication conditions.

On the other hand, if it is marked "N" (it means "No") as shown in Figure 1, i.e., the upper limit of time-delay is equal to one sampling period, in other words the possibility of allowing more transmission time-delay to keep system stable exists (the larger delay can be addressed as $0 < \tau_k < lh, l > 1$, where $\tau_k$ is the time-delay and $h$ is the sampling period). In this case, compared with augmented state vector for time-delay less than one sampling period in system model, which may be defined as $z(kh) = \left[ x^T(kh), u^T((k-1)h) \right]^T$, the state vector becomes more augmented, $z(kh) = \left[ x^T(kh), u^T((k-l)h) \dots, u^T((k-1)h) \right]^T$. Meanwhile, one question related to larger time-delay will be asked: How much delay the system can tolerate? Moreover, more allowable time-delay will facilitate the reconciliation of communication efficiency and transmission preciseness. In this analytical methodology, stability analyzer and time-delay boundary estimations will have contributions to continually release more network resources. As seen in Figure 1, system with larger or smaller time-delay has their own processing method in this methodology. They can be collected to provide the design rules for network-based control system in power electronic area, and the flow direction is addressed with red arrow in Figure 1.

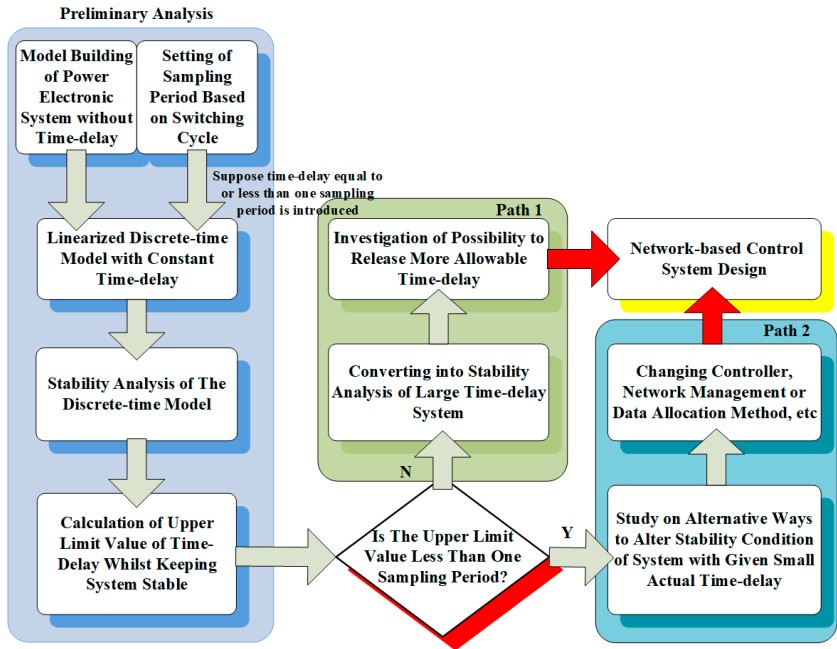

**Figure 1.** Flow diagram of one analytical methodology for evaluating impact of bounded time-delay on power electronic system stability and performance with the basic time unit being switching period.

If the time-delay introduced by network is time-invariable and then static scheduling network protocols are envisioned (such as token ring or token bus) [30], the model can be simplified to form

the discrete time-invariable function. In the proposed analytical methodology, switching period being the basic time unit for analysis can facilitate the implementation of network-based control on power electronic system, which is originally designed by digital controller with possible sampling equal to switching cycle.

## 3. Network-Based Control Methods for Inverter Parallel Operation

To characterize the merits of network-based control used in power electronic system, a number of inverters that can operate in parallel to share heavy loads and communication link, which can be equivalently regarded as islanded mode in AC microgrids, are constructed, and addressed in Figure 2. Information data of the inverters can share each other through network link. The network protocol should be designed by the feature of transmission data and Medium Access Control (MAC) of network [30].

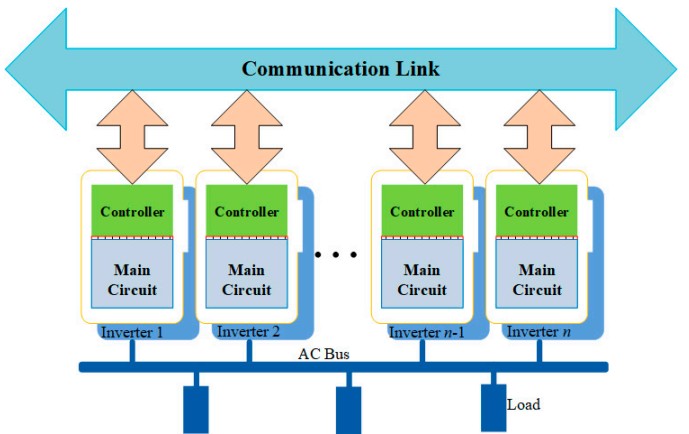

**Figure 2.** Network-based control system for parallel operation inverters.

The AC/DC inverters operating in parallel mode are equivalent to voltage source inverters (VSIs). The controller includes voltage/current control loops and droop control method. If the link impedance and output impedance are considered inductive and the equivalent circuit for two inverters in parallel are shown in Figure 3, the active and reactive power injected to the bus by every unit are expressed as follows [31]:

$$P_n = \frac{E_n U}{X_n} \sin \phi_n, \tag{1}$$

$$Q_n = \frac{E_n U \cos \phi_n - U^2}{X_n}, \tag{2}$$

where $E_n$ and $U$ are the magnitudes of the nth inverter output voltage and the common bus voltage respectively, $\phi_n$ is the power angle, and the sum of output impedance and link impedance is $Z_n = r_n + jX_n = R_{Zn} \angle \theta_n$, where $r_n$ is the equivalent resistance and $X_n$ is the equivalent inductive reactance for the $n^{th}$ inverter.

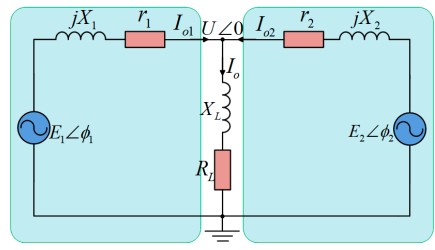

**Figure 3.** Equivalent circuit of two inverters in parallel operation.

Considering a small phase difference between $E_n$ and $U$, it can be seen from Equations (1) and (2) that the active $P_n$ is strongly dependent on the power angle $\phi_n$, while the reactive power is mainly influenced by the amplitude difference $E_n$-$U$ [31]. Consequently, most wireless load sharing controllers introduce artificial droops into the output voltage reference. The frequency and the amplitude of the inverter output-voltage reference can be expressed as

$$\begin{cases} \omega_n = \omega^* - k_{p\omega}P_n \\ E_n = E^* - k_{qE}Q_n \end{cases} \tag{3}$$

When the difference of sum of output impedance and link impedance in the two inverters is small, the circulating current can be eliminated by balancing the active and the reactive power. Therefore, we choose active power $P$ and reactive power $Q$ as the transmission signals in the network-based control system. In the following section, two current-sharing strategies based on droop control are presented, the difference between them is mainly reflected as to the extent to which current-sharing performance can address the impact of time-delay.

It is essential to calculate power variables $P_n/Q_n$ before completing droop control. In general, they are obtained from instantaneous power passing through low-pass filters (LPF) with a smaller bandwidth than that of the closed-loop inverter. Due to the existence of different power, two network-based control strategies for inverter parallel systems are proposed and indicated in Figure 4, labeled as Steady Power Method (SPM) and Transient Power Method (TPM), respectively. From the view of transmission mode, the communication is conducted by every inverter and there is no relationship for them as clearly defined as master/slave (autonomous mode).

### 3.1. Steady Power Method (SPM)

The Steady Power Method (SPM) is shown in Figure 4a, and the two-inverter parallel system is used to illustrate its principle. The power transmission over network are averaged one after passing through LPF for conquering disturbs. The control power $P_n/Q_n$ are obtained by creating a weighted form of its own data and communication data, presented as follows

$$\text{ForInverter\#1,} \quad \left. \begin{array}{c} P_1 = m_A P_{2\_d} + (1 - m_A)P_{1\_c} \\ Q_1 = n_A Q_{2\_d} + (1 - n_A)Q_{1\_c} \\ m_A \leq 1, n_A \leq 1 \end{array} \right\}, \tag{4}$$

$$\text{ForInverter\#2,} \quad \left. \begin{array}{c} P_2 = m_B P_{1\_d} + (1 - m_B)P_{2\_c} \\ Q_2 = n_B Q_{1\_d} + (1 - n_B)Q_{2\_c} \\ m_B \leq 1, n_B \leq 1 \end{array} \right\}. \tag{5}$$

Assume that $n$ = 1,2 represents the number of the inverter, $m_A$, $n_A$, $m_B$, and $n_B$ are weighted coefficient for indicating the proportion of network data participating in control of each paralleled inverter, which are all available for autonomous regulation and amplitude limited within 1.0. For example, $m_A$ represents the ratio of network-obtaining active power $P_{2\_d}$ of inverter #2 in control active power $P_1$ of inverter #1. $P_n$, $Q_n$ are the final active and reactive power for droop control, $P_{n\_d}$, $Q_{n\_d}$ are the power data received from network, and $P_{n\_c}$, $Q_{n\_c}$ are the calculated power on-line derived from output current and voltage of inverter, respectively. The droop control is implemented by each inverter and followed under control function (3).

From Figure 4a, we can deduce that if there are no time-delays and data dropouts during transmission, i.e., lossless network, it should have

$$\left. \begin{array}{c} P_{n\_c} = P_{n\_d} \\ Q_{n\_c} = Q_{n\_d} \end{array} \right\} \tag{6}$$

The data flow on either side of the network follows the directions arrow with the same color, which is marked by pink and yellow color in Figure 4a.

## 3.2. Transient Power Method (TPM)

Another method, the Transient Power Method (TPM), with the example of the two-inverter parallel system, is shown in Figure 4b, where each inverter has two kinds of transient powers (active and reactive power) transferred to communication bus. The transient power is obtained from the calculation based on sampling output current and voltage of the inverters. Once each inverter obtains the power data of others from network, a weighted process towards output instantaneous active and reactive power is activated and served for obtaining control power by Low-Pass Filters (LPF). This weighted function is defined as

$$\text{For Inverter\#1,} \quad \left.\begin{array}{l} p_1 = m_C p_{2\_d} + (1 - m_C)p_{1\_c} \\ q_1 = n_C q_{2\_d} + (1 - n_C)q_{1\_c} \\ m_C \le 1, n_C \le 1 \end{array}\right\} \tag{7}$$

$$\text{For Inverter\#2,} \quad \left.\begin{array}{l} p_2 = m_D p_{1\_d} + (1 - m_D)p_{2\_c} \\ q_2 = n_D q_{1\_d} + (1 - n_D)q_{2\_c} \\ m_D \le 1, n_D \le 1 \end{array}\right\}. \tag{8}$$

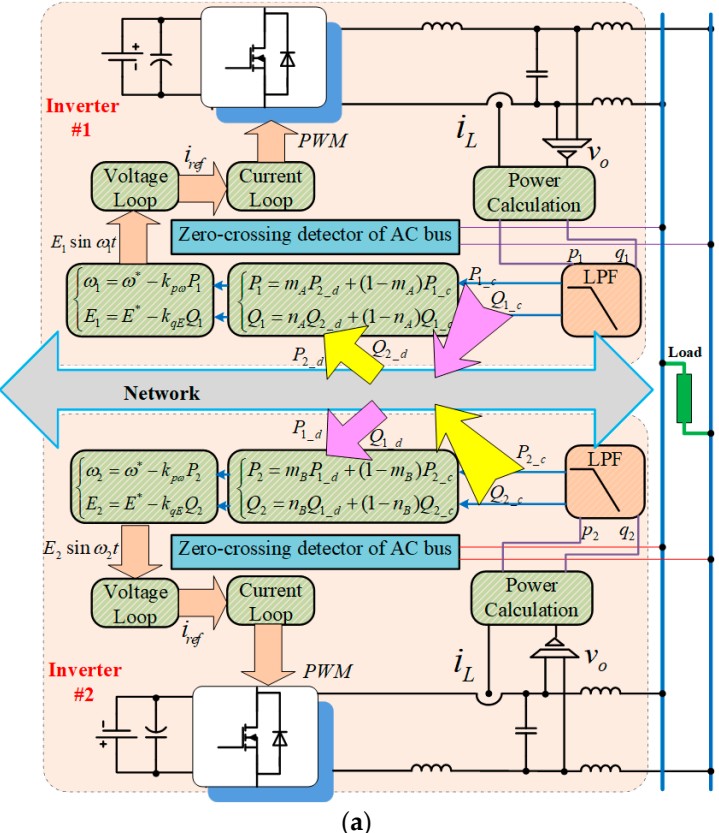

(**a**)

**Figure 4.** *Cont.*

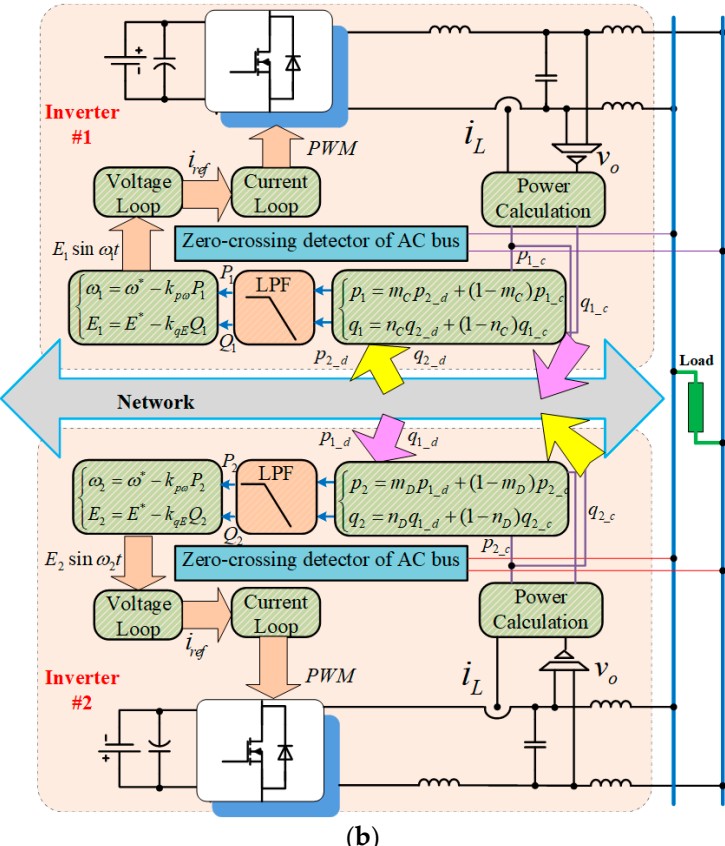

**(b)**

**Figure 4.** Two methods of network-based control system for parallel operation with two inverters. (**a**) Steady Power Method (SPM), (**b**) Transient Power Method (TPM).

The new $p_n/q_n$ (n = 1, 2 are the sequence number of the inverter) are the weighted results of transient active and reactive power of each inverter, $m_C$, $n_C$, $m_D$, and $n_D$ are the weighted coefficients for describing the proportion of network data in power control, $p_{n\_c}$, $q_{n\_c}$ are the power calculated by sampling voltage and current for each inverter. Simultaneously, they are transmitted via network and received by other inverter by labeling by $p_{n\_d}$ and $q_{n\_d}$. If on-line network transmission is successful, it will have

$$\left.\begin{array}{l} p_{n\_c} = p_{n\_d} \\ q_{n\_c} = q_{n\_d} \end{array}\right\} \tag{9}$$

The weighted results of $p_n$ and $q_n$ are sent to LPF to get averaged power $P_n$ and $Q_n$, respectively, i.e., the ultimate power in droop control function (3). The data flow direction from one inverter to another via network is highlighted with pink and yellow as well.

### 3.3. Preliminary Analysis: Sensitivity of Time-Delay for Two Methods

Based on the description in Section 2, whether the communication-induced time-delay is larger or smaller than one sampling period, the basic stability of system within one sampling period could be investigated, which had been marked as "Preliminary Analysis" in Figure 1. A linearized discrete-time model is adopted to describe these two strategies, their state space functions are addressed by (A5) and (A9) in the Appendix A.

Considering the case of time-delay of each sample, $\tau_k$, being less than one sampling period $h$, the system equation can be written as

$$
\begin{aligned}
\dot{x}(t) &= Ax(t) + Bu(t), \quad t \in [kh + \tau_k, (k+1)h + \tau_{k+1}), \\
y(t) &= Cx(t), \\
u(t^+) &= -Kx(t - \tau_k), \quad t \in \{kh + \tau_k, k = 0, 1, 2, \ldots\}
\end{aligned}
\tag{10}
$$

where $u(t^+)$ is piecewise continuous and only changes its value at $kh + \tau_k$. When the system is sampled with period $h$, it can be obtained by

$$
\begin{aligned}
x((k+1)h) &= \Phi x(kh) + \Gamma_0(\tau_k)u(kh) + \Gamma_1(\tau_k)u((k-1)h), \\
y(kh) &= Cx(kh)
\end{aligned}
\tag{11}
$$

where $\Phi = e^{Ah}$, $\Gamma_0(\tau_k) = \int_0^{h-\tau_k} e^{As} B ds$, $\Gamma_1(\tau_k) = \int_{h-\tau_k}^{h} e^{As} B ds$.

Defining $z(kh) = \left[x^T(kh), x^T((k-1)h)\right]^T$ as the augmented state vector, the augmented closed-loop system is [32],

$$
z((k+1)h) = \widetilde{\Phi}(k)z(kh),
\tag{12}
$$

where

$$
\widetilde{\Phi}(k) = \begin{bmatrix} \Phi - \Gamma_0(\tau_k)K & \Gamma_1(\tau_k) \\ -K & 0 \end{bmatrix}.
\tag{13}
$$

In this case, we can use the exponential stability criterion, to judge whether $\widetilde{\Phi}(\tau)$ is a stable Schur matrix. To explain the "Preliminary Analysis" in the proposed methodology with the example statement and investigate the sensitivity of time-delay on these two methods, the implementation schemes are provided below.

Test for SPM: (1) Average power filtered by LPF is transmitted through CAN bus and the sampling period is 150 $\mu s$; (2) the link impedance and output impedance are inductive; (3) to simplify the control analysis, we suppose the control coefficients satisfying $m_A = m_B = m_S = 0.3$, $n_A = n_B = n_S = 0.6$.

Test for TPM: (1) Instantaneous power is transmitted through CAN bus and the power sample period is 150 $\mu s$; (2) the link impedance and output impedance are inductive; (3) the predefined control coefficients $m_C = m_D = m_T = 0.2$, $n_C = n_D = n_T = 0.3$.

The main parameters are presented, and the specification for paralleled inverters is: $R_L = 50\ \Omega$, $X_L = 0.02\ \Omega$, $X_1 = X_2 = 0.628\ \Omega$, $\vec{E}_1 = (216.7 + j38)V$, $\vec{E}_2 = (216.7 + j35)V$, and droop control coefficients are: $k_{p\omega} = 7.5 \times 10 - 4 (\text{rad}/(\text{W·s}))$, $k_{qE} = 3.5 \times 10 - 3 (\text{V}/\text{Var})$.

Analysis methodology discussed in Section 2 tells us that no matter how much the actual time-delay is, the rough estimation of stability within one sampling time could be carried out. Returned to the strategies proposed in Section 3, stability analysis becomes multidimensional due to the existence of control coefficient $m_S$, $n_S$, $m_T$, $n_T$, and time-delay $\tau_k$. The boundary curves used for addressing the upper limit time-delay maintaining system still stable are obtained by Schur criterion, as shown in Figure 5. The result reveals that the stability features of SPM and TPM towards time-delay are remarkably different. In SPM result as depicted in Figure 5a, where three-dimension coordinates are adjustable coefficients $m_s$, $n_s$ and maximum allowable time-delay, there are few cross-columns with different depths to show that only these regions with particular $m_s/n_s$ combinations have relatively large allowable time-delay, the rest area is covered by zero-close flat surface. It means network-based system stability is dominantly affected by time-delay in SPM. Nevertheless, in the TPM results shown in Figure 5b, except only a small part of region showing the allowable time-delay is tiny, other upper value in the regional surface can be rushed into 150 $\mu s$, which could be even bigger if we release the 150 $\mu s$ restriction. The conclusion can be drawn that system is less influential on the time-delay impact in TPM. From the angle of control, the system shows its susceptibility to the time-delay and the control parameters should be elaborately designed for $m_S$ and $n_S$ to keep system stable in SPM. On the other hand, control coefficient $m_T$ and $n_T$ still cannot be designed arbitrarily even system is basically regarded as delay-insensitive in TPM. In addition, the stability criteria described in this section is used to reach a guidance conclusion on the qualitative analysis of time-delay impact. The quantitative

accuracy is dependent on the precise model of power electronic systems and the theoretical analysis methods we are seeking.

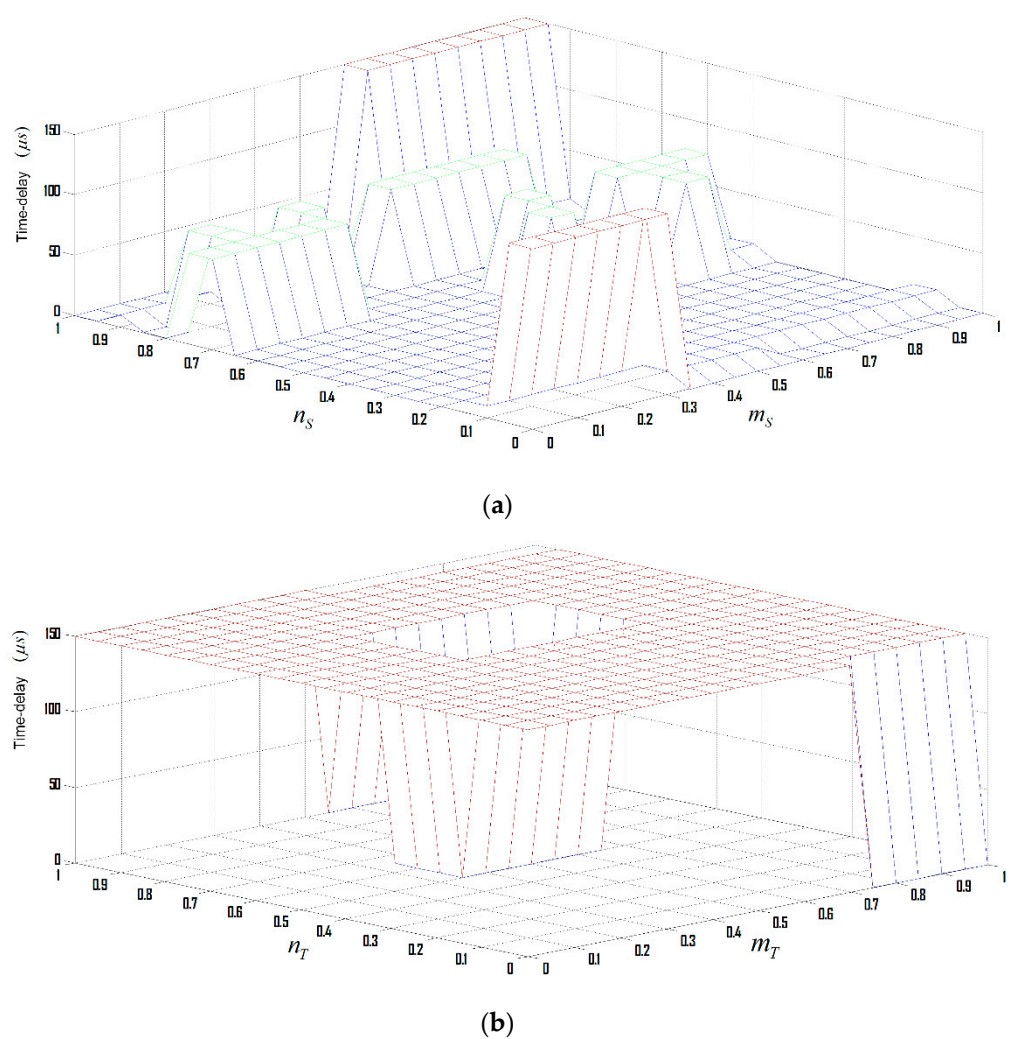

**Figure 5.** Upper limit value of time-delay whilst keeping system stable for different $m_S$, $n_S$, $m_T$, and $n_T$. (**a**) SPM. (**b**) TPM.

## 4. Two Analytical Paths to Handle Time-Delay Impact

Since time-delay could possibly generate the negative errors for the control system and may severely damage system performance or stability, the feasible way is to eliminate or shorten the inherent delay but it's possibly impracticable. Therefore, there are two paths (Path 1 and Path 2, framed in Figure 1) that can be traced when confronting the impacts of time-delay guided by the proposed methodology in Section 2. The first path involves exploring the maximum time-delay while preserving satisfactory performance so that more network capacity could be released. The transmission load would be decreased and correspondingly the system could be optimized in cost and reliability. In addition, when we find the network-based system is highly sensitive to the time-delay, besides network reselection and improved data-processing method [16], redesigning the controller in accordance with specific hardware or network conditions is another effective plan. The main contribution of "Preliminary Analysis" is to classify the method into delay-insensitive and delay-sensitive strategy according to their delay attribute. In this section, "Path 1" and "Path 2" addressed in Section 2 are theoretically analyzed, with the concrete verification by two current-sharing network-based strategies proposed in Section 3.

### 4.1. "Path 1" For Delay-Insensitive Strategy

Due to the different susceptibility to inherent time-delay during communication, the network-based control method can be categorized into delay-insensitive and delay-sensitive one. Based on the description above, TPM can be more viewed as delay-insensitive strategy. Because the maximum time-delay considered in "Preliminary Analysis" is limited to one sampling period, there is a substantial possibility for this kind of delay-insensitive system to have a larger allowable time-delay (maybe many time sampling periods) to ensure system stable.

As discussed in Section 1, the network-based control tends to be a practical and easy-to-accomplish policy with prioritization of using the existing network tools. However, when it is employed in power electronic systems, the transmitted signals become complex and the required communication rate will be on an incredibly wide range. This is one reason why we define switching cycle as basic unit of sampling period, which was emphasized in Section 2. The problems arising from the increasing communication collision would degrade the performance of power electronic systems and reduce the reliability. Meanwhile, we hope the desired transmission rate for control signals could be less than we expect so that more network resources can be released, so there exist a compromise between performance improvement and increase of allowable time-delay.

The "Path 1" gives one potential solution for delay-insensitive strategy. Since the allowable time-delay may exceed one sampling period, the desired performance under a relatively low transmission rate could be obtained. In this case the possibility of releasing more network load needs to be theoretically investigated, which offers greater potential under the circumstance of power electronic systems with high switching frequency. The stability analysis based on model-building will be converted into the one with large time-delay. One following example will illustrate how to estimate the quantity of allowable time-delay a stable system can tolerate.

Suppose the network-based control has one-channel feedback, variable (like inductor current $i_L$ and output voltage $v_o$ as shown in Figure 4) sampling is running in time-driven pattern, which is easy to be acquired by digital controller. In addition, controller and PWM (Pulse Width Modulation) drive can be in either time-driven or event-driven operation, mostly dependent on the communication mode and control design. If we define time reference as the data arrival time at PWM drive, the close network-based system can be described as

$$\begin{cases} \dot{x}(t) = Ax(t) + Bu(t), t \in [i_k h + \tau_k, i_{k+1} h + \tau_{k+1}) \\ u(t^+) = Kx(t - \tau_k), t \in \{i_k h + \tau_k, k = 1, 2, \ldots\} \end{cases}, \qquad (14)$$

where $h$ is sampling period, $i_k$ ($k = 1,2,3, \ldots$) is positive integer and $\{i_1, i_2, i_3, \ldots\} \subset \{0, 1, 2, \ldots\}$, $\tau_k$ represents the time-consuming for data No. $i_{kh}$ passing from sampling point to PWM drive, apparently there is a relationship as follows

$$\overset{\infty}{\underset{k=1}{\cup}} [i_k h + \tau_k, i_{k+1} h + \tau_{k+1}) = [t_0, \infty), t_0 \geq 0 \qquad (15)$$

Assume $u(t) = 0$ before the first control signal reaching PWM driven terminator. There is no data package dropout happens during no-loss transmission in proper order, $\{i_1, i_2, i_3, \ldots\} = \{0,1,2, \ldots\}$. When $i_{k+1} < i_k$, it means out-of-sequence transmission happening and the controller is considered event-driven, the system can be rewritten as

$$\begin{cases} \dot{x}(t) = Ax(t) + BKx(i_k h), & t \in [i_k h + \tau_k, i_{k+1} h + \tau_{k+1}) \\ x(t) = x(t_0 - \eta)e^{A(t - t_0 + \eta)}, & t \in [t_0 - \eta, t_0] \end{cases}, \qquad (16)$$

where $\eta$ is upper limit value of $\{(i_{k+1} - i_k)h + \tau_{k+1}, k = 1, 2, \ldots\}$. Equation (16) can be designed in network-based system with both time-varying and constant delays. To simplify the analysis and explain "Path 1" visually, $\tau_k$ is defined as having bounded and non-constant attributes.

The stability analysis of system (14,16) is used to seek maximum time-delay to achieve stability and satisfactory performance. There are, of course, numerous mathematical methods that can be applied and stability criteria as well. Take one example, as for system (16), there are two alternative conditions are addressed as follows.

Definition 1: Suppose there exists constant $\alpha > 0$ and $\beta > 0$ such that the solution for system (16) solution satisfies, $\|x(t)\| \leq \alpha \sup_{t_0 - \eta \leq s \leq t_0} \|\phi(s)\| e^{-\beta t}, t \geq t_0$ the system (16) is exponentially asymptotically stable.

Criterion 1 [33]: For a given scalar $\eta > 0$, suppose that there exist square matrices $P, Mi, Ni(I = 1,2,3)$ and positive definite matrix $T > 0$ such that (17) holds. Then, the network-based control system (16) is asymptotically stable as long as $(i_{k+1} - i_k)h + \tau_{k+1} \leq \eta, k = 1,2,3, \ldots,$

$$
\begin{bmatrix}
N_1 + N_1^T - M_1 A - A^T M_1^T & N_2^T - N_1 - A^T M_2^T - M_1 BK \\
* & -N_2 - N_2^T - M_2 BK - K^T B^T M_2^T \\
* & \\
* & * \\
N_3^T - A^T M_3^T + M_1 + P & \eta N_1 \\
-M_3^T + M_2 - K^T B^T M_3^T & \eta N_2 \\
M_3 + M_3^T + \eta T & \eta N_3 \\
* & -\eta T
\end{bmatrix} < 0 \tag{17}
$$

where * is symmetric item of matrix.

As network-based control system with TPM is considered, $A$, $B$ and $K$ are derived from state-space function (A9).

## 4.2. "Path 2" For Delay-Insensitive Strategy

Based on "Preliminary Analysis," "Path 1" is created for delay-insensitive strategy to achieve the balance between performance and network burden, as outlined in Section 2. Nevertheless, more increased efforts need to be done to enhance the stability and performance of delay-sensitive strategy with considering the vulnerability features of network. In addition, it is necessary to investigate how to minimize the performance impact of the network with fast transmission speed, which is mostly built based on high frequency characteristics in power electronic system. "Path 2" framed in Figure 1 provides one creative direction on modifying the system based on small time-consuming network conditions. The reformed ways can be changing controller, network management or data allocation method, etc. However, there are few studies on these currently, especially in power electronic field. Therefore, as an exploratory research, one possible theoretical solution for delay-sensitive strategy, where control coefficients and gain are redesigned based on the bounded and random time-delay, is introduced.

In general, the mathematical model and statistical disciplinarian of time-delay need to be known in advance, giving it mathematical representation, such as Poisson Process, Markov chain and ARMA (Autoregressive moving average) model, and so on. However, in actual network environment, time-delay always exists in random style. As discussed in Section 3, there is the greater possibility to have the stability issues when transmission collision is increased, and time-delay feature is unknown, which it is normal for network-based control used in high-frequency power electronic system. Narrowed at network-based control strategies shown in Section 3, for given control coefficients $m_T$, $n_T$, essentially feedback gain $K$, the system is very likely to be unstable due to its delay-sensitive characteristics. Therefore, our purpose is to design a controller such that the unstable system is changeable. and the performance can be enhanced. Hereby the improvement of delay-sensitive strategy is started from the change of controller according to the time-delay attribute of network.

One possible way to analyze system with unpredictable time-delay is to converter generalized discrete-time model for controlled object into some types of linear discrete model with uncertainty, so that many existing mathematical tools can be available for use.

The stability of such a system (A22) having the measurable state variables is given by one of theorems.

Criterion 2: Assume that there exists symmetric positive definite matrix $P$, $Q$, scalar $\varepsilon > 0$, the network-based control system (16) with controller of $u(t^+) = Kx(t - \tau_k)$ is exponentially stable if there exist feedback gain matrix $K$ such that

$$
\begin{bmatrix}
\varepsilon HH^T - P^{-1} & e^{Ah} + B_0 K & B_1 K & 0 \\
(e^{Ah} + B_0 K)^T & -P + Q & 0 & (FK)^T \\
(B_1 K)^T & 0 & -Q & (-FK)^T \\
0 & FK & -FK & -\varepsilon I
\end{bmatrix} < 0.
\tag{18}
$$

Through using MATLAB (MathWorks, Natick, MA, USA) and its Linear Matrix Inequality (LMI) toolbox to solve (18), we can deduce the calculation results based on the specification described in Section 3, $B_0$, $B_1$, $H$, and $F$ are obtained in matrices (A24–A27). Feedback gain matrix is modified to be $K$, as rewritten in (A28).

## 5. Experimental Results

To demonstrate the validity of proposed analysis methodology using in parallel operation of inverters, two 220 Vac/1kVA 50 Hz inverters were built. DSP (Digital Signal Processing) was employed, and the control coefficients of droop control and current/voltage double-loop were delicately designed. The general structure of network-based control system with two inverters in parallel using SPM and TPM, the reference voltage of each inverter is yielded from droop control and then plunged into current control.

In both SPM and TPM, the active and reactive power signals outputting from each inverter can be transmitted via CAN bus in autonomous and equal style instead of master-slave mode. Time-stamp was specifically designed for identifying transmission order by setting ID with numerical increment during transmission period. In addition, zero-crossing detector of AC bus was employed as auxiliary function to help ID synchronization in communication. In addition, it also plays the role of synchronizer for the initial action of parallel operation. For each inverter, their CAN controller was responsible for receiving and sending power information $p_{n\_c}$, $P_{n\_c}$ along with specified ID. Meanwhile, the buffer inside CAN controller can be used to store data and manufacture sending time for transmission, which facilitate the time-delay test by artificially setting different transmission time.

### 5.1. For Delay-Insensitive Strategy (Transient Power Method)

To test the impact of time-delay on SPM, experiments were designed with given control coefficients of $m_A = m_B = 0.2$ and $n_A = n_B = 0.3$. The mathematical results obtained from Section 3 were based on the premise that time-delay is equal to or less than one sampling period (150 $\mu s$ here), it showed that TPM is immune to time-delay of at least one sampling period. This is experimentally validated by output current of parallel inverters, which was demonstrated in Figure 6. As we can see, good current-sharing performance is achieved in two different loads. To further describe its delay-insensitive feature and relatively good function, the current-sharing error is first introduced by $|(I_1 - (I_1 + I_2)/2)/((I_1 + I_2)/2)| \times 100\%$, where $I_1$ and $I_2$ are simultaneously recorded as output current RMS of each paralleled inverter, which can be obtained by oscilloscope or multimeter. In the test, 30 groups of simultaneously captured $I_1$ and $I_2$, which each adjacent group was measured at 5-min intervals during steady-state operation for parallel, were used to calculate 30 current-sharing errors. Average error and maximum error were numerically processed into statistical data. The former is to get the averaging value and the latter is to find the maximum value from these 30 errors. Similar

data-processing method was adopted in droop-only-method (which is represented by Equation (3)), so that the average and maximum current-sharing errors can be compared between these two methods, the results were addressed in Figure 7. It is clearly shown that in most occasion, both average and maximum current-sharing error achieved by using Transient Power Method are less than those with droop control method without network. More valuably, these errors can be strictly restricted within 5% even though the proposed method has to tolerate 150 $\mu s$ inherent time-delay for transmission, which demonstrates the delay-insensitive feature and feasibility of network-based control.

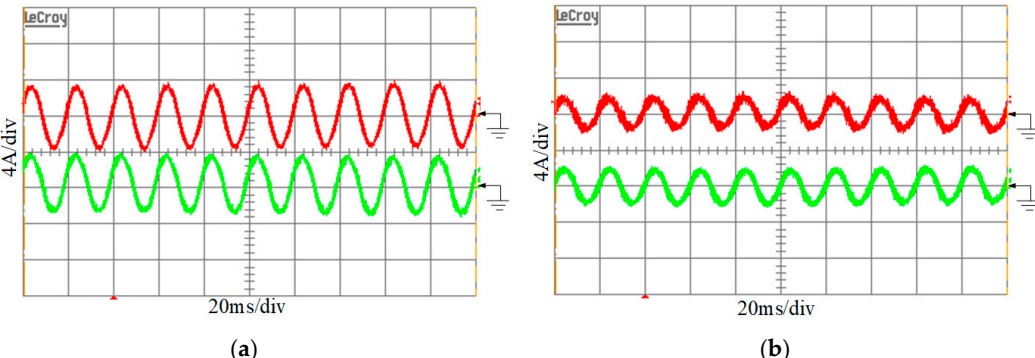

(**a**)　　　　　　　　　　　　　　　　　　　　　　　　　　　(**b**)

**Figure 6.** Output current of parallel system by using TPM in two different resistance loads. (**a**) $R_L$ = 50 Ω. (**b**) $R_L$ = 100 Ω.

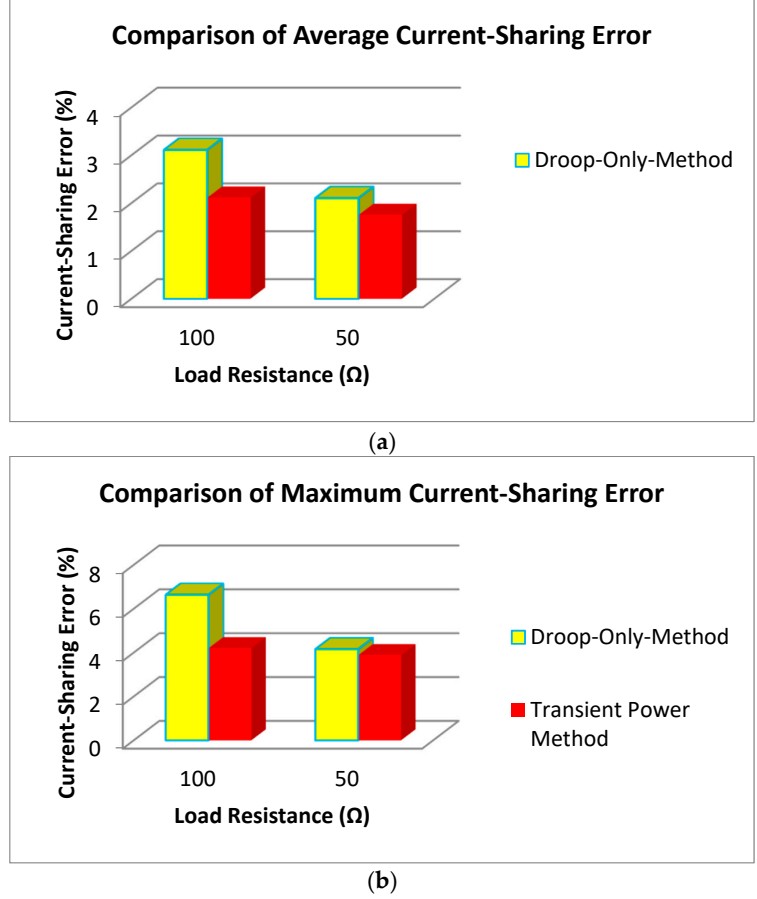

**Figure 7.** Comparison of average and maximum current-sharing errors by using droop-only-method and TPM. (**a**) Average current-sharing error. (**b**) Maximum current-sharing error.

Because of its delay-insensitive characteristics, "Path 1" is proposed to aim at reaching a compromise between system performance and network load release, as depicted in Section 2. Now, we can deduce that TPM is delay-insensitive, and capable of tolerating 150 μs network-induced delay for most coefficient area ($m_A$, $m_B$, $n_A$, and $n_B$ area). However, we do not know its potential for releasing network capacity. One possible way, as "Path 1" discussed, is to achieve maximum allowable time-delay while keeping system work in desired performance by mathematical analysis and experimental verification.

The current sampling $i_L$, voltage sampling $v_o$ and PWM output were set to be time-driven and synchronous mode, which is easy for DSP implementation. Time-delay can be simulated as having bounded and non-constant attribute. Controller is equipped with event-driven mode, which means control is activated when receiving/sending communication signals. For convenience sake, time-delay was artificially set as bounded to test its system impact. The system performance including its stability condition can be observed through output current of parallel inverters with variant $\eta_{max}$, being shown in Figure 8. As seen, when the delays increase, the output current-sharing of inverter parallel system rapidly degrades. When $\eta_{max}$ is randomly chosen to be 450 μs (Figure 8a), good performance is still maintained. But when $\eta_{max}$ is up to 750 μs (that means there are three data packets discarded between two adjacent controlled power signals when sampling period is 150 μs), system exhibits the tendency of instability, as shown in Figure 8b. With more $\eta_{max}$, a worse current-sharing performance will be obtained. $\eta_{max}$ =900 μs is leading to system in instability trajectory (see Figure 8c). When $\eta_{max}$ is up to 3 *ms* (Figure 8d), when the second inverter takes part in parallel operation, over-current protection is active and turns off the whole system, in other words the system becomes collapse. The experimental results roughly fit the mathematical values offered in Section 3.

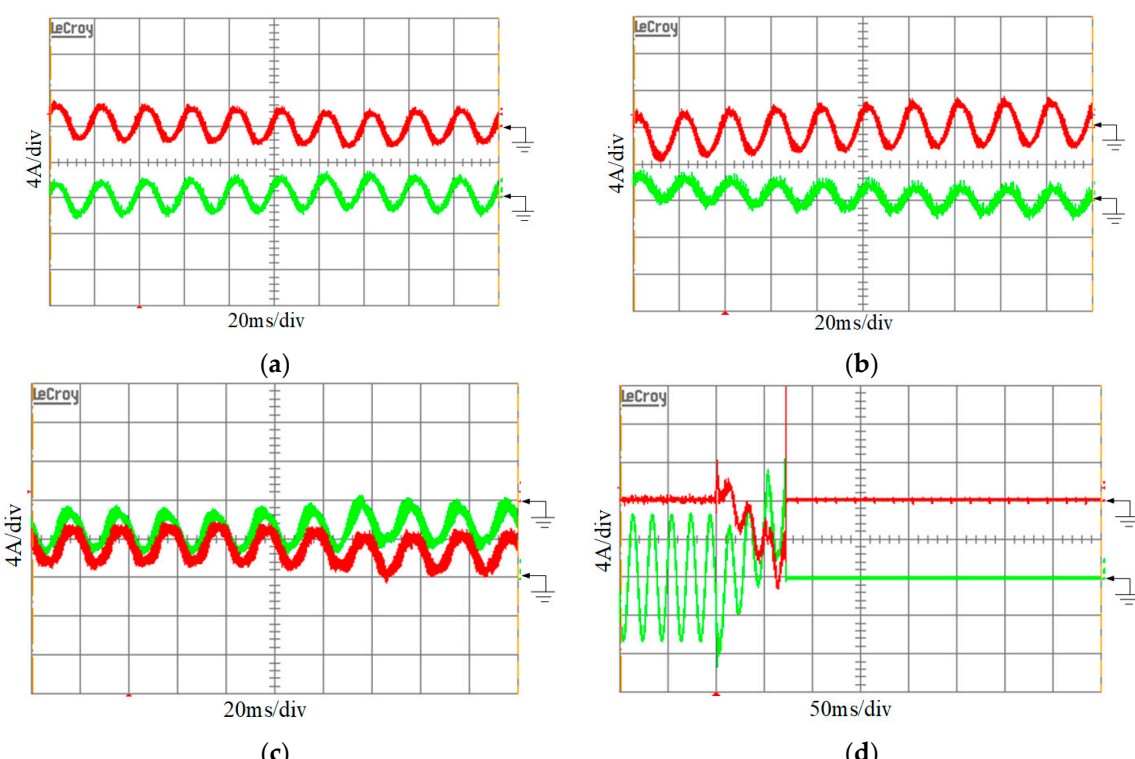

**Figure 8.** Output current of parallel system with network-based control in different $\eta_{max}$. (**a**) $\eta_{max}$ = 450 μs. (**b**) $\eta_{max}$ = 750 μs. (**c**) $\eta_{max}$ = 900 μs. (**d**) $\eta_{max}$ = 3 *ms*.

*5.2. For Delay-Insensitive Strategy (SPM)*

For SPM as discussed in Section 3, its weak resistivity over time-delay accelerates its speed to pursuit reliable and high-speed network. Based on the analysis methodology proposed in Section 2,

"Path 2" offers one practical way to reverse original system instability and enhance the performance if we keep the network condition unchanged. In this experimental test, control coefficients were predefined as $m_S = 0.3$, $n_S = 0.6$, the output current of parallel inverter system is shown in Figure 9a, dynamical property is not desirable, and it takes long time to reach relatively steady state. In addition, the system seems to be unstable even after several oscillation periods. From Figure 9a, it can be seen that output current is highly distorted and current-sharing error is considerably large in steady-state. In this scenario, based on Criterion 2, the control feedback gain matrix $K$, which is contained by $m_A$, $m_B$, $n_A$ and $n_B$, is modified, and a new $K$ illustrated by (A28) is achieved. Corresponding change can be reflected by experimental results, which was demonstrated by Figure 9b. In both the dynamic and steady state, the parallel system witnessed significant improvements. It needs to be noted that "Path 2" contains a number of methods besides altering controller based on time-delay property. For example, the change of data-processing method of transmission to enhance performance, which had been tried in Reference [16]. Deviations exist between theoretical and experimental verification toward modified control feedback, and how to reduce and eliminate them is mostly dependent upon accuracy of mathematical model building on power electronic system.

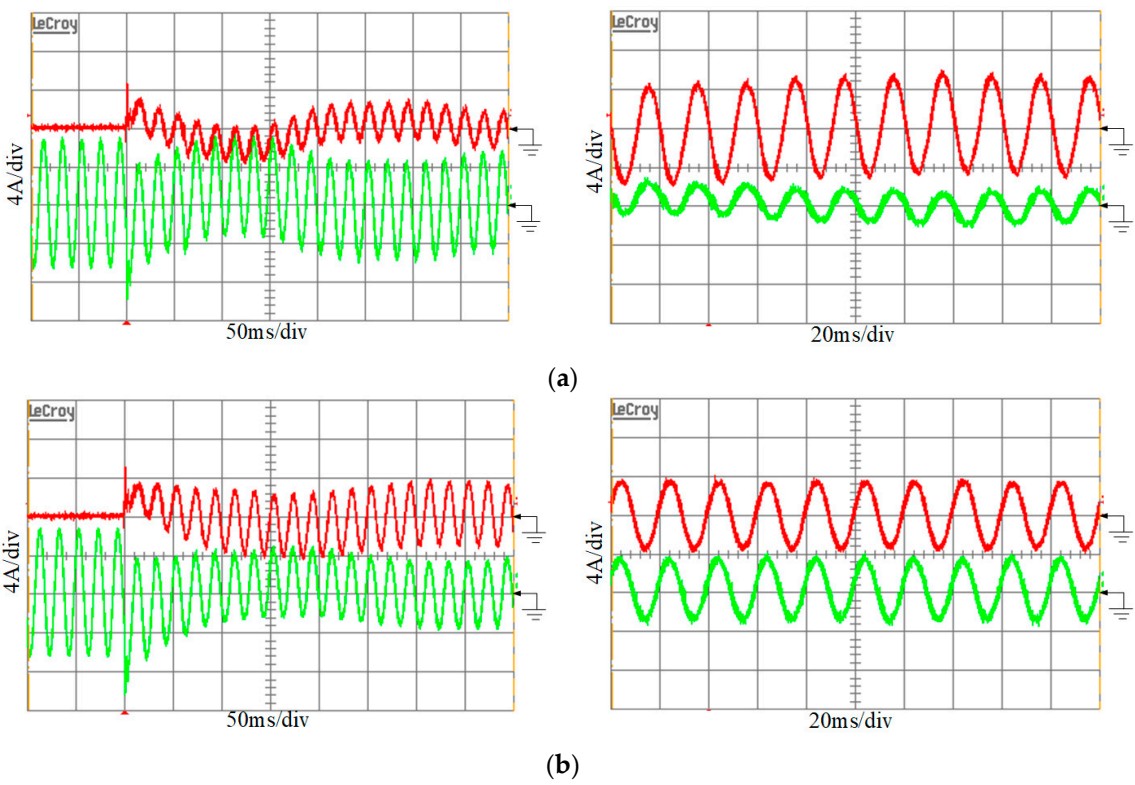

**Figure 9.** Output current of parallel system with SPM. (**a**) Using original $K'$. (**b**) Modified $K$.

## 6. Conclusions

Two current-sharing with network-based control strategies for droop-based AC microgrid in islanded mode are proposed in this paper. Due to the inherent existence of time-delay during network transmission, each strategy needs to be investigated its impact on system stability and performance. Accordingly, a series of theoretical methodology for analyzing how to handle time-delay in network-based control of power electronic systems was proposed. The basic idea is to build mathematical model having sampling period equal to multiples of switching cycle and to theoretically test system stability within one sampling period, in which the delay-sensitive and delay-insensitive feature are roughly distinguished. To obtain the optimized performance and reduce the possibility of instability without adding communication complexity and cost, two paths are proposed. As to

delay-insensitive method, the main idea of "Path 1" in analysis methodology, is to reduce the transmission frequency and the occupancy of network while maintaining system in satisfied operation performance. On the other hand, "Path 2" points out there are variant ways to improve performance in delay-sensitive strategy, including changing of network management, data allocation method, etc., although less efforts had been done previously. In this paper, the redesign of controller in regard to special network condition was concerned. The analysis methodology is employed in two current-sharing strategies of inverter-parallel operation in islanded mode of AC microgrids, concrete implementation and analytical methods are presented as well. It provides guidance and support for improving performance of power electronic systems, expands the usage of network-based control for power electronic field in a manner that is itself highly reliable, efficient, and managed.

**Funding:** This research was funded by the National Natural Science Foundation of China under grant 51877058, in part by Natural Science Foundation of Zhejiang Province under grant LY18E070004.

**Conflicts of Interest:** The author declare no conflict of interest. The funders had no role in the design of the study; in the collections, analyses, or interpretation of data; in the writing of the manuscript, or in the decision to publish the results.

## Appendix A

*Appendix A.1. Modeling of Two Network-based Control Methods*

The mathematical model is built based on the applied equivalent circuit shown in Figure 3. Make an assumption: $X_1 = X_2 = X$, $r_1 = r_2 = r$, the transient active and reactive power of two inverters around the equilibrium point $(\phi_{1e}, \phi_{2e}, E_{1e}, E_{2e})$ could be expressed as [16],

$$\begin{bmatrix} \dot{p}'_1 \\ \dot{p}'_2 \\ \dot{q}'_1 \\ \dot{q}'_2 \end{bmatrix} = S \begin{bmatrix} \dot{\phi}_1 \\ \dot{\phi}_2 \\ \dot{E}_1 \\ \dot{E}_2 \end{bmatrix} = S \begin{bmatrix} \omega_1 \\ \omega_2 \\ \dot{E}_1 \\ \dot{E}_2 \end{bmatrix}, S = [s_{ij}], i, j = 1 \dots 4, \tag{A1}$$

Appendix A.1.1. SPM

LPF (low-pass filtering) indicated in Figure 4 is used to remove the noise of power data, we create $\omega_f$ as its cut-off angular frequency and the relational formula between filter result $P$ and filter input $p$ is,

$$P = \frac{\omega_f}{s + \omega_f} p. \tag{A2}$$

If the communication action is successful, all data is transmitted without time-delay and data dropout, the weighted functions of power are defined as,

$$\begin{aligned} P_{W1} &= m_A P_{2\_c} + (1 - m_A) P_{1\_c}, \\ P_{W2} &= m_B P_{1\_c} + (1 - m_B) P_{2\_c}, \\ Q_{W1} &= n_A Q_{2\_c} + (1 - n_A) Q_{1\_c}, \\ Q_{W2} &= n_B Q_{1\_c} + (1 - n_B) Q_{2\_c}. \end{aligned} \tag{A3}$$

$P_{Wn}, Q_{Wn}$ are then substituted into droop control function (3). Define the following states vector:

$$\begin{cases} x = \begin{bmatrix} \dot{P}_{1\_c} & \dot{P}_{2\_c} & \dot{Q}_{1\_c} & \dot{Q}_{2\_c} & P_{1\_c} & P_{2\_c} \end{bmatrix}^T \\ u = \begin{bmatrix} \dot{P}_{W1} & \dot{P}_{W2} & \dot{Q}_{W1} & \dot{Q}_{W2} & P_{W1} & P_{W2} \end{bmatrix}^T \end{cases}, \tag{A4}$$

Equations (A1), (A2), (A3) and (3) are combined and represented as:

$$\begin{cases} \dot{x}(t) = Ax(t) + Bu(t) \\ u(t) = Kx(t) \end{cases},$$ (A5)

where $u(t)$ is control variable and $K$ is the feedback gain and

$$A = \begin{bmatrix} -\omega_f & 0 & -s_{13}k_{qE}\omega_f & 0 & -s_{11}k_{p\omega}\omega_f & 0 \\ 0 & -\omega_f & -s_{23}k_{qE}\omega_f & 0 & -s_{21}k_{p\omega}\omega_f & 0 \\ 0 & 0 & -s_{33}k_{qE}\omega_f - \omega_f & 0 & -s_{31}k_{p\omega}\omega_f & 0 \\ 0 & 0 & -s_{43}k_{qE}\omega_f & -\omega_f & -s_{41}k_{p\omega}\omega_f & 0 \\ 1 & 0 & 0 & 0 & 0 & 0 \\ 0 & 1 & 0 & 0 & 0 & 0 \end{bmatrix},$$

$$B = \begin{bmatrix} 0 & 0 & 0 & -s_{14}k_{qE}\omega_f & 0 & -s_{12}k_{p\omega}\omega_f \\ 0 & 0 & 0 & -s_{24}k_{qE}\omega_f & 0 & -s_{22}k_{p\omega}\omega_f \\ 0 & 0 & 0 & -s_{34}k_{qE}\omega_f & 0 & -s_{32}k_{p\omega}\omega_f \\ 0 & 0 & 0 & -s_{44}k_{qE}\omega_f & 0 & -s_{42}k_{p\omega}\omega_f \\ 0 & 0 & 0 & 0 & 0 & 0 \\ 0 & 0 & 0 & 0 & 0 & 0 \end{bmatrix},$$ (A6)

$$K = \begin{bmatrix} 1 - m_A & m_A & 0 & 0 & 0 & 0 \\ m_B & 1 - m_B & 0 & 0 & 0 & 0 \\ 0 & 0 & 1 - n_A & n_A & 0 & 0 \\ 0 & 0 & n_B & 1 - n_B & 0 & 0 \\ 0 & 0 & 0 & 0 & m_A & 1 - m_A \\ 0 & 0 & 0 & 0 & m_B & 1 - m_B \end{bmatrix}.$$

Appendix A.1.2. TPM

The model of TPM can be likewise deduced and fully presented.

Suppose the transmission of power data among inverters is completely successfully and without time-delay cost, the weighted functions of transient power are defined as,

$$\begin{aligned} p_{W1} &= m_C p_{2\_c} + (1 - m_C)p_{1\_c}, \\ p_{W2} &= m_D p_{1\_c} + (1 - m_D)p_{2\_c}, \\ q_{W1} &= n_C q_{2\_c} + (1 - n_C)q_{1\_c}, \\ q_{W2} &= n_D q_{1\_c} + (1 - n_D)q_{2\_c}. \end{aligned}$$ (A7)

The transient power $p_{Wn}$, $q_{Wn}$ obtained from (A7) pass through LPF to get filtered power, which can be substituted into droop control function (3).

Define the following states vector:

$$\begin{cases} x = \left[\omega_1 - \omega_1^* \, \omega_2^* - \omega_2 E_1^* - E_1 E_2^* - E_2 p_{1\_c} \, p_{2\_c} \, q_{1\_c} \, q_{2\_c}\right]^T \\ u = \left[\omega_1 - \omega_1^* \, \omega_2^* - \omega_2 E_1^* - E_1 E_2^* - E_2 p_{W1} \, p_{W2} \, q_{W1} \, q_{W2}\right]^T \end{cases},$$ (A8)

System model of TPM is addressed as:

$$\begin{cases} \dot{x}(t) = Ax(t) + Bu(t) \\ u(t) = Kx(t) \end{cases},$$ (A9)

where u(t) is control variable and $K$ is the feedback gain and,

$$
A = \begin{bmatrix}
-\omega_f & 0 & 0 & 0 & -\omega_f k_{p\omega} & 0 & 0 & 0 \\
0 & -\omega_f & 0 & 0 & 0 & 0 & 0 & 0 \\
0 & 0 & -\omega_f & 0 & 0 & 0 & \omega_f k_{qE} & 0 \\
0 & 0 & 0 & -\omega_f & 0 & 0 & 0 & 0 \\
s_{11} & -s_{12} & s_{13}\omega_f & s_{14}\omega_f & 0 & 0 & -s_{13}\omega_f k_{qE} & 0 \\
s_{21} & -s_{22} & s_{23}\omega_f & s_{24}\omega_f & 0 & 0 & -s_{23}\omega_f k_{qE} & 0 \\
s_{31} & -s_{32} & s_{33}\omega_f & s_{34}\omega_f & 0 & 0 & -s_{33}\omega_f k_{qE} & 0 \\
s_{41} & -s_{42} & s_{43}\omega_f & s_{44}\omega_f & 0 & 0 & -s_{43}\omega_f k_{qE} & 0
\end{bmatrix},
$$

$$
B = \begin{bmatrix}
0 & 0 & 0 & 0 & 0 & 0 & 0 & 0 \\
0 & 0 & 0 & 0 & 0 & \omega_f k_{p\omega} & 0 & 0 \\
0 & 0 & 0 & 0 & 0 & 0 & 0 & 0 \\
0 & 0 & 0 & 0 & 0 & 0 & 0 & \omega_f k_{qE} \\
0 & 0 & 0 & 0 & 0 & 0 & 0 & -s_{14}\omega_f k_{qE} \\
0 & 0 & 0 & 0 & 0 & 0 & 0 & -s_{24}\omega_f k_{qE} \\
0 & 0 & 0 & 0 & 0 & 0 & 0 & -s_{34}\omega_f k_{qE} \\
0 & 0 & 0 & 0 & 0 & 0 & 0 & -s_{44}\omega_f k_{qE}
\end{bmatrix},
\qquad \text{(A10)}
$$

$$
K = \begin{bmatrix}
1 & 0 & 0 & 0 & 0 & 0 & 0 & 0 \\
0 & 1 & 0 & 0 & 0 & 0 & 0 & 0 \\
0 & 0 & 1 & 0 & 0 & 0 & 0 & 0 \\
0 & 0 & 0 & 1 & 0 & 0 & 0 & 0 \\
0 & 0 & 0 & 0 & 1-m_C & m_C & 0 & 0 \\
0 & 0 & 0 & 0 & m_D & 1-m_D & 0 & 0 \\
0 & 0 & 0 & 0 & 0 & 0 & n_C & 1-n_C \\
0 & 0 & 0 & 0 & 0 & 0 & n_D & 1-n_D
\end{bmatrix},
$$

Joint the models of two methods, $p_{Wn}$, $q_{Wn}$, $P_{Wn}$, $Q_{Wn}$ are the ultimate power obtained by weighted functions (A3) and (A7), $p_{n\_c}$, $q_{n\_c}$, $P_{n\_c}$, $Q_{n\_c}$ are the active and reactive power of each inverter in on-line calculation, $m_A$, $n_A$, $m_B$, $n_B$, $m_C$, $n_C$, $m_D$ and $n_D$ are the weighted coefficient for network-based control. Equation (A5) and (A9) describe the system stability around the equilibrium point and $A$ is the characteristic matrix.

*Appendix A.2. Modeling of Network-based Control System with Time-varying Time-delay*

In this case, data dropout is neglected and network-based control system (11) is built with the presence of time-delay $\tau_k$. Considering its time-varying feature, $\Gamma_0(\tau_k)$ and $\Gamma_1(\tau_k)$ are time-varying too. Function (11) needs to be transformed in convenient for redesigning controller and the work is started with studying characteristic matrix $A$.

Scenario 1: $A$ has n different characteristic roots.

$A$ can be converted into diagonal matrix, and rewritten as $A = \Lambda diag(\lambda_1, \ldots, \lambda_n)\Lambda^{-1}$ by means of diagonal matrix transformation, where $\lambda_1, \ldots, \lambda_n$ are the characteristic roots of $A$. $\Lambda = [\Lambda_1, \ldots, \Lambda_n]$ is matrix formed by characteristic roots of $A$. Then it has,

$$
\begin{aligned}
\Gamma_0(\tau_k)B &= \int_0^{h-\tau_k} e^{At}dt B = \Lambda diag\left(\int_0^{h-\tau_k} e^{\lambda_1 t}dt, \ldots, \int_0^{h-\tau_k} e^{\lambda_n t}dt\right)\Lambda^{-1}B \\
&= B_0 + \Lambda diag\left(\tfrac{1}{\lambda_1}e^{\lambda_1(h-\tau_k)}, \ldots, \tfrac{1}{\lambda_n}e^{\lambda_n(h-\tau_k)}\right)\Lambda^{-1}B
\end{aligned}
\qquad \text{(A11)}
$$

where

$$
B_0 = \Lambda diag\left(-\tfrac{1}{\lambda_1}, \ldots, -\tfrac{1}{\lambda_n}\right)\Lambda^{-1}B
$$
$$
\begin{aligned}
\Lambda diag&\left(\tfrac{1}{\lambda_1}e^{\lambda_1(h-\tau_k)}, \ldots, \tfrac{1}{\lambda_n}e^{\lambda_n(h-\tau_k)}\right)\Lambda^{-1}B \\
&= \Lambda diag\left(\tfrac{1}{\lambda_1}e^{\lambda_1\beta_1}, \ldots, \tfrac{1}{\lambda_n}e^{\lambda_n\beta_n}\right) \times diag\left(\tfrac{1}{\lambda_1}e^{\lambda_1(h-\beta_1-\tau_k)}, \ldots, \tfrac{1}{\lambda_n}e^{\lambda_n(h-\beta_n-\tau_k)}\right)\Lambda^{-1}B \\
&= HM(\tau_k)F
\end{aligned}
\qquad \text{(A12)}
$$

and,

$$H = \Lambda diag(\frac{1}{\lambda_1}e^{\lambda_1\beta_1}, \ldots, \frac{1}{\lambda_n}e^{\lambda_n\beta_n}) \tag{A13}$$

$$M(\tau_k) = (\frac{1}{\lambda_1}e^{\lambda_1(h-\beta_1-\tau_k)}, \ldots, \frac{1}{\lambda_n}e^{\lambda_n(h-\beta_n-\tau_k)}) \tag{A14}$$

$$F = \Lambda^{-1}B \tag{A15}$$

wherein $e^{\lambda_1(h-\beta_i-\tau_k)} < 1, i = 1, \ldots, n$.

It can be obtained by using the same method,

$$\Gamma_1(\tau_k)B = B_1 - HM(\tau_k)F \tag{A16}$$

and $B_1 = \Lambda diag(-\frac{1}{\lambda_1}e^{\lambda_1 h}, \ldots, -\frac{1}{\lambda_n}e^{\lambda_n h})\Lambda^{-1}B$.

Scenario 2: $A$ has characteristic roots consisting of a zero, $r$ multiple root $\lambda^*$ and other nonzero distinctly different roots.

In this scenario, $A$ can be rewritten as $A = \Lambda diag(0, J_1, J_2)\Lambda^{-1}$, where $J_1$ is diagonal matrix composed by $\lambda_2, \ldots, \lambda_{n-r}$, $J_2 \in R^{r \times r}$ is the Jordan block with its corresponding $\lambda^*$. The result from similar calculation is,

$$B_0 = \Lambda diag(h, \widehat{J}_1, \widehat{J}_2)\Lambda^{-1}B, \tag{A17}$$

where

$$\widehat{J}_1 = \begin{bmatrix} -(1/\lambda_2) & & \\ & \ddots & \\ & & -(1/\lambda_{n-r}) \end{bmatrix}; \widehat{J}_2 = \begin{cases} \begin{bmatrix} -\frac{1}{\lambda^*} & & & \\ \frac{1}{\lambda^{*2}} & \ddots & & \\ \vdots & \ddots & \ddots & \\ (-1)^r\frac{1}{\lambda^{*r}} & \cdots & \frac{1}{\lambda^{*2}} & -\frac{1}{\lambda^*} \end{bmatrix}, \lambda^* \neq 0 \\ diag(h, \ldots, h), \qquad\qquad \lambda^* = 0, \end{cases} \tag{A18}$$

$$B_1 = \Lambda diag(0, \breve{J}_1, \breve{J}_2)\Lambda^{-1}B$$

and therein,

$$\breve{J}_1 = \begin{bmatrix} (1/\lambda_2)e^{\lambda_2 h} & & \\ & \ddots & \\ & & (1/\lambda_{n-r})e^{\lambda_{n-r}h} \end{bmatrix};$$

$$\breve{J}_2 = \begin{cases} \begin{bmatrix} -\frac{1}{\lambda^*}e^{\lambda^* h} & & & \\ \frac{1}{\lambda^{*2}}(\lambda^* h - 1)e^{\lambda^* h} & \ddots & & \\ \vdots & \ddots & \ddots & \\ \frac{1}{\lambda^{*r}}\sum_{k=1}^{r}(-1)^{k-1}\frac{(\lambda^* h)^{r-k}}{(r-k)!}e^{\lambda^* h} & \cdots & \frac{1}{\lambda^{*2}}(\lambda^* h - 1)e^{\lambda^* h} & -\frac{1}{\lambda^*}e^{\lambda^* h} \end{bmatrix}, \lambda^* \neq 0 \\ \begin{bmatrix} 0 & & & 0 \\ T^2/2 & \ddots & & \\ \vdots & \ddots & \ddots & \\ T^r/r! & \cdots & T^2/2 & 0 \end{bmatrix}, \qquad\qquad \lambda^* = 0, \end{cases} \tag{A19}$$

$$H = \Lambda diag(\alpha_1, \frac{1}{\lambda_2}e^{\lambda_2\beta_2}, \cdots, \frac{1}{\lambda_{n-r}}e^{\lambda_{n-r}\beta_{n-r}}, P_2)$$

where $\alpha_1 > \tau_k$, choose $\alpha_l, l = 2, \cdots, n - r$ such that $e^{\lambda_1(T-\tau_k-\alpha_l)} < 1$. $P_2 \in \mathbb{R}^{r \times r}$ is diagonal and invertible matrix, making $\|P_2^{-1}\widetilde{J}_2\| < 1$ satisfied. Therein,

$$\widetilde{J}_2 = \begin{cases} \begin{bmatrix} -\frac{1}{\lambda^*}e^{\lambda^*(h-\tau_k)} & & & \\ \frac{1}{\lambda^{*2}}(\lambda^*(h-\tau_k)-1)e^{\lambda^*(h-\tau_k)} & \ddots & & \\ \vdots & & \ddots & \ddots \\ \frac{1}{\lambda^{*r}}\sum_{j=1}^{r}(-1)^{j-1}\frac{[\lambda^*(h-\tau_k)]^{r-j}}{(r-j)!}e^{\lambda^*(h-\tau_k)} & \cdots & \frac{1}{\lambda^{*2}}(\lambda^*(h-\tau_k)-1)e^{\lambda^*(h-\tau_k)} & -\frac{1}{\lambda^*}e^{\lambda^*(h-\tau_k)} \end{bmatrix}, \lambda^* \neq 0 \\[4em] \begin{bmatrix} -\tau_k & & & 0 \\ (h-\tau_k)^2/2 & \ddots & & \\ \vdots & & \ddots & \ddots \\ (h-\tau_k)^r/r! & \cdots & (h-\tau_k)^2/2 & -\tau_k \end{bmatrix}, \quad \lambda^* = 0, \end{cases}$$  (A20)

$$M(\tau_k) = diag(-\tfrac{\tau_k}{\beta_1}, e^{\lambda_2(h-\tau_k-\beta_2)}, \cdots, e^{\lambda_{n-r}(h-\tau_k-\beta_{n-r})}, P_2^{-1}\widetilde{J}_2)$$

$$F = \Lambda^{-1}B$$  (A21)

Now, the discrete time model of generalized controlled object can be presented as,

$$x((k+1)h) = \Phi x(kh) + (B_0 + HM(\tau_k)F)u(kh) + (B_1 - HM(\tau_k)F)u((k-1)h)$$  (A22)

$B_0$, $B_1$, $H$ and $F$ are time-invariant matrices, and $M^T(\tau_k)M(\tau_k) \leq I$, $M(\tau_k)$ is addressed as $M$.

In addition, suppose there is a single-channel control feedback loop, i. e., $u(k) = K\hat{x}(k)$, where $K \in \mathbb{R}^{1 \times n}$ is feedback gain constant matrix and $\hat{x}(k)$ is the input of the controller, (A22) can be transformed into,

$$x((k+1)h) = \Phi x(kh) + (B_0 + HMFK)x(kh) + (B_1K - HMFK)x((k-1)h)$$  (A23)

Given the specification of network-based control system mentioned in Section 3, correlated matrices can be deduced as,

$$B_0 = \begin{bmatrix} 0 & 0 & 0 & -0.0015 & 0 & 4.7503 \\ 0 & 0 & 0 & -0.0051 & 0 & -4.7516 \\ 0 & 0 & 0 & 0.2405 & 0 & -0.0198 \\ 0 & 0 & 0 & -0.2405 & 0 & 0.0198 \\ 0 & 0 & 0 & 0 & 0 & -0.0101 \\ 0 & 0 & 0 & -0.0014 & 0 & 0.0101 \end{bmatrix}$$  (A24)

$$B_1 = \begin{bmatrix} 0 & 0 & 0 & 0.0015 & 0 & -4.6803 \\ 0 & 0 & 0 & 0.0051 & 0 & 4.6816 \\ 0 & 0 & 0 & -0.2370 & 0 & 0.0203 \\ 0 & 0 & 0 & 0.2369 & 0 & -0.0202 \\ 0 & 0 & 0 & 0 & 0 & 0.0101 \\ 0 & 0 & 0 & 0.0014 & 0 & -0.0101 \end{bmatrix}$$  (A25)

$$H = \begin{bmatrix} 0 & 0 & 0 & 0.0055 & -0.707 & 0.0230 \\ 0 & 0 & -0.0104 & -0.0055 & 0.7071 & 0.0125 \\ 0 & 0 & 0 & 0 & 0 & 1.4142 \\ 0.0001 & 0 & 0 & 0 & 0 & -1.4140 \\ 0 & 0 & 0 & -0.0011 & 0.0074 & -0.0002 \\ 0 & -0.2021 & 0.0001 & 0.0011 & -0.0074 & -0.0001 \end{bmatrix}$$  (A26)

$$
F = \begin{bmatrix}
0 & 0 & 0 & -0.0076 & 0 & 0 \\
0 & 0 & 0 & -0.0072 & 0 & 0 \\
0 & 0 & 0 & -1.2083 & 0 & 0 \\
0 & 0 & 0 & -0.0192 & 0 & 37.2275 \\
0 & 0 & 0 & 0.7860 & 0 & -70.13488 \\
0 & 0 & 0 & 34.0106 & 0 & 3.6069
\end{bmatrix}
\tag{A27}
$$

Based on Criterion 2, the modified control gain can be finally obtained through predefinition of some special matrices *P*, *Q*,

$$
K = \begin{bmatrix}
0.85 & 0.15 & 0 & 0 & 0 & 0 \\
0.25 & 0.75 & 0 & 0 & 0 & 0 \\
0 & 0 & 0.247 & 0.753 & 0 & 0 \\
0 & 0 & 0.442 & 0.558 & 0 & 0 \\
0 & 0 & 0 & 0 & 0.15 & 0.85 \\
0 & 0 & 0 & 0 & 0.25 & 0.75
\end{bmatrix}
\tag{A28}
$$

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
