# Peer review of "Analysis Methodology for Evaluation of Time-Delay Impact on Network-Based System for Droop-Controlled AC Microgrid"

_electronics, doi:10.3390/electronics8040380_

Round 1

Reviewer 1 Report

The contribution is interesting, the research approach is visible. However it is recommended to improve the paper abstract e.g. try to be more specific in terms of the paper novelty, some general information can be given in the paper introduction. The sentence like “It is well-known” shall be omitted. The results presented in Fig. 5 shall be better described.

Author Response

Point 1: The contribution is interesting, the research approach is visible. However it is recommended to improve the paper abstract e.g. try to be more specific in terms of the paper novelty, some general information can be given in the paper introduction. The sentence like “It is well-known” shall be omitted. The results presented in Fig. 5 shall be better described.

Response 1: Thank you very much. You are right. The abstract should be more explicit in terms of main idea and novelty of the paper. Abstract was rewritten, and more general information was added in the introduction, as shown in the revised version. The supplemental description of Figure 5 was given as well.

Reviewer 2 Report

Dear Author,

I have some comments to your article:

1.       Please check the abstract carefully and enter the modifications so that it corresponds fully with the content of the article.

2.       All indexes in symbols in text and equations should be checked carefully.

3.       Line 95 - resources [12][24] - a comma should be used to separate the numbers of the cited literature ([12], [24]).

4.       Lines 152-153  Meanwhile, there is question related to larger time-delay we cannot avoid, “How much delay can the system tolerate?” Without personal forms. Similarly, line 270. Please check the entire text carefully.

5.       My fears about the authorship of the wording "we" author is only one. Also section - Conflicts of Interest: The authors declare no conflict of interest.

6.       A poorly described diagram shown in Figure 1.

7.       Line 187  - some word before rn

8.       Line 195 - no colon before the equation, check for other equations.

9.       The adopted coefficients have not been sufficiently explained in the text - mA, nA, mB and nB are weighted coefficient.

10.   Transient Power Method, - you could use the acronym TPM, similarly Steady Power Method - SPM.

11.   Line 260 does not need one comma

12.   lines 455 - 460 please provide more details about the measurement method and the measuring equipment used.

13.   It is necessary to more thoroughly check the database of articles and it is advisable to cite especially the literature items from the last 18 months.

Author Response

Point 1: Please check the abstract carefully and enter the modifications so that it corresponds fully with the content of the article.

Response 1: Thanks for your suggestion. The abstract was rewritten to give more explicit information so that it can correspond fully with the content of this paper.

Point 2: All indexes in symbols in text and equations should be checked carefully.

Response 2: Thanks for your suggestion. The indexes had been checked and corrected in the revised version.

Point 3: Line 95 - resources [12][24] - a comma should be used to separate the numbers of the cited literature ([12], [24]).

Response 3: Thanks for your comments. It was corrected in the revised version.

Point 4: Lines 152-153  Meanwhile, there is question related to larger time-delay we cannot avoid, “How much delay can the system tolerate?” Without personal forms. Similarly, line 270. Please check the entire text carefully.

Response 4: Thank you very much. I had checked the entire text carefully and revised them.

Point 5: My fears about the authorship of the wording "we" author is only one. Also section - Conflicts of Interest: The authors declare no conflict of interest.

Response 5: Thanks for your comments. This paper is an independent completion since I have no student to assist in current university. I believe things will be changed soon in the future and more contributors will join in my work and articles. Because the author is only me and there is no “authors” in this paper, I deleted the section-Conflicts of Interest to avoid controversy.

Point 6: A poorly described diagram shown in Figure 1.

Response 6: Thank you very much. The description of Figure 1 had been completed in Section 2.

Point 7: Line 187  - some word before rn

Response 7: Thanks for your comments. “where” was added before rn.

Point 8: Line 195 - no colon before the equation, check for other equations.

Response 8: Thank you very much. The same problems had been checked and corrected in the revised version.

Point 9: The adopted coefficients have not been sufficiently explained in the text - mA, nA, mB and nB are weighted coefficient.

Response 9: Thanks for your comments. The supplemental description of these coefficients was added in the revised version.

Point 10: Transient Power Method, - you could use the acronym TPM, similarly Steady Power Method - SPM.

Response 10: Many thanks. Based on your suggestions, the acronyms were used in this paper.

Point 11: Line 260 does not need one comma

Response 11: Thank you very much. I had deleted the comma in the revised version.

Point 12: lines 455 - 460 please provide more details about the measurement method and the measuring equipment used.

Response 12: Thanks for your comments. The description was completed in lines 455-460.

Point 13: It is necessary to more thoroughly check the database of articles and it is advisable to cite especially the literature items from the last 18 months.

Response 13: Thanks for your suggestions. I thoroughly checked the database of articles and supplemented some references in the revised version.